# Sensitivity of atmospheric aerosol scavenging to precipitation intensity and frequency in the context of global climate change

Pei Hou [1, 2], Shiliang Wu [1, 2, 3], Jessica L. McCarty [4, 5], Yang Gao[3]

[1]Atmospheric Sciences Program, Michigan Technological University, Houghton, MI, 49931, USA.
[2]Dept. of Geological and Mining Engineering and Sciences, Michigan Technological University, Houghton, MI, 49931, USA.
[3]College of Environmental Science and Engineering, Ocean University of China, Qingdao, China.
[4]Michigan Tech Research Institute, Ann Arbor, MI, 48105, USA.
[5]Dept. of Geography, Miami University, Oxford, OH, 45056, USA.

*Correspondence to*: Shiliang Wu (slwu@mtu.edu)

**Abstract.** Wet deposition driven by precipitation is an important sink for atmospheric aerosols and soluble gases. We investigate the sensitivity of atmospheric aerosol lifetimes to precipitation intensity and frequency in the context of global climate change. Our sensitivity model simulations, through some simplified perturbations to precipitation in the GEOS-Chem model, show that the removal efficiency and hence the atmospheric lifetime of aerosols have significantly higher sensitivities to precipitation frequencies than to precipitation intensities, indicating that the same amount of precipitation may lead to different removal efficiencies of atmospheric aerosols. Combining the long-term trends of precipitation patterns for various regions with the sensitivities of atmospheric aerosol lifetimes to various precipitation characteristics allows us to examine the potential impacts of precipitation changes on atmospheric aerosols. Analyses based on an observational dataset show that precipitation frequency in some regions have decreased in the past 14 years, which might increase the atmospheric aerosol lifetimes in those regions. Similar analyses based on multiple reanalysis meteorological datasets indicate that the changes of precipitation intensity and frequency over the past 30 years can lead to perturbations in the atmospheric aerosol lifetimes by 10% or higher at the regional scale.

## 1 Introduction

Wet scavenging is a major removal process for aerosols and soluble trace gases [*Atlas and Giam*, 1988; *Radke et al.*, 1980]. Global climate change implies significant perturbations of precipitation, which can directly affect the wet scavenging process. *Salzmann* [2016] found that the global mean precipitation did

not change significantly since 1850 with climate models, while *Trenberth et al.* [2007] reported that the total precipitation amount increased over land north of 30°N in the past century and decreased in the tropical region after the 1970s based on observational data. *Trenberth* [2011] also noted that theoretically a warmer climate could lead to less frequent but more intense precipitation.

The impacts of long-term changes in precipitation characteristics on air quality have not been well studied. Most previous studies focused on the correlation between air pollution and the total precipitation amount or precipitation intensity [*Cape et al.*, 2012; *Pye et al.*, 2009; *Tai et al.*, 2012]. For example, *Dawson et al.* [2007] found a strong sensitivity of the $PM_{2.5}$ (particulate matter with diameter less than 2.5 µm) concentrations to precipitation intensity over a large domain of the eastern US with perturbation tests. Only a few studies focused on precipitation frequency. *Jacob and Winner* [2009] noted that precipitation frequency could be more important than precipitation intensity for air quality, because the wet scavenging process due to precipitation is very efficient [*Balkanski et al.*, 1993]. *Fang et al.* [2011] projected with the Geophysical Fluid Dynamics Laboratory chemistry-climate model (AM3) that wet deposition has a stronger spatial correlation with precipitation frequency than intensity over the US in January, although they concluded that frequency has a minor effect on wet deposition in the context of climate change. *Mahowald et al.* [2011] also discussed the importance of precipitation frequency in wet deposition based on simulations showing large removal rate of dust in precipitation events.

In this study, we first use GEOS-Chem, a global 3-D chemical transport model (CTM), to examine the sensitivities of atmospheric aerosol lifetimes to various precipitation characteristics, including the precipitation intensity, frequency, and total amount. By isolating these precipitation characteristics from other meteorological fields through a suite of perturbation simulations, we are able to better understand the sensitivities of atmospheric aerosols to various precipitation characteristics. We focus on black carbon (BC) as a proxy for atmospheric aerosols to examine the impacts of changes in precipitation characteristics. BC is nearly inert in the atmosphere [*Ramanathan and Carmichael*, 2008], making it a good tracer for studying the transport and deposition of atmospheric species. We also analyze the long-term trends of the precipitation characteristics over various regions around the world, based on the observational and reanalysis meteorological datasets for the past decades. We then combine the long-term trends in the precipitation patterns for various regions with the sensitivities of BC to

precipitation characteristics to quantify their potential impacts on atmospheric aerosols in the context of global climate change.

**2 Methods**

We utilize a global 3-D chemical transport model (CTM), GEOS-Chem [*Bey et al.*, 2001] version 9-02-01 (www.geos-chem.org), to carry out a suite of perturbation tests to examine the sensitivities of atmospheric aerosols to precipitation characteristics. As a chemical transport model, the GEOS-Chem model does not simulate meteorology prognostically; instead it is driven by assimilated meteorological data from the Goddard Earth Observing System (GEOS) of NASA GMAO. We use the GEOS-5 meteorological dataset in this study. We conduct global simulations with a horizontal resolution of 4° latitude by 5° longitude and 47 vertical layers. All the model simulations in this study run from 1 July 2005 to 1 January 2007, i.e., for one and half years, with the first half year serving as the model spin-up. The wet deposition scheme in GEOS-Chem includes scavenging in convective updrafts, in-cloud scavenging (rainout), and below-cloud scavenging (washout), which were described in detail by *Liu et al.* [2001] and *Wang* et al. [2011]. In GEOS-Chem simulation, the BC aerosols are classified into two types based on their hygroscopicity (hydrophobic vs hydrophilic) and wet scavenging is more efficient for hydrophilic BC. GEOS-Chem assumes the ratio between hydrophobic and hydrophilic BC to be 4:1 in fresh emissions and hydrophobic BC converts to hydrophilic one with an e-folding lifetime of 1.15 days.

The washout rate constant ($k$) is affected by the particle size and the form of precipitation. For washout by rain with precipitation rate $P$ (mm/h), $k = 1.1 \times 10^{-3} P^{0.61}$ for accumulation mode (aerosols with diameters between 0.04 μm and 2.5 μm) and $k = 0.92 P^{0.79}$ for coarse mode (aerosols with diameter between 2.5 μm to 16 μm); for washout by snow with precipitation rate $P$, $k = 2.8 \times 10^{-2} P^{0.96}$ for accumulation mode and $k = 1.57 P^{0.96}$ for coarse mode [*Feng, 2007, 2009*]. The coefficients for accumulation-mode are used in calculating k for fine particles including BC in GEOS-Chem.

Our study focuses on three precipitation characteristics: the precipitation intensity, frequency, and total amount. We define precipitation events as the data points with "significant" (we use precipitation rate more than 1 mm/day as the criterion in this study) precipitation. Precipitation intensity is the average

precipitation rate on precipitation events, with a unit of mm/day. Precipitation frequency is the fraction of precipitation events during the study period (i.e., the probability of any given data points with more than 1 mm/day precipitation rate), which is dimensionless. Total precipitation amount is defined as the average amount of precipitation rate during the study period, with a unit of mm/day. Assuming that precipitation is negligible on data points with no "precipitation events", we would have

$$total\ precipitation\ amount \cong precipitation\ intensity \cdot precipitation\ frequency \qquad (1)$$

For sensitivity tests focused on precipitation intensity, we scale the base GEOS-5 precipitation values from the control run by a uniform factor for each grid box. For the sensitivity tests focused on precipitation frequency, we use a stochastic function to turn off the precipitation at a given data point. For example, in a simulation where we reduce the precipitation frequency by 25%, for a data point $(i, j, t)$, we modify the initial precipitation rate $P_0(i, j, t)$ to

$$P(i, j, t) = \begin{cases} P_0(i, j, t); & R(i, j, t) \geq 0.25 \\ 0; & R(i, j, t) < 0.25 \end{cases} \qquad (2)$$

where $R$ is a random function with a range of (0, 1). In this way, we decrease the precipitation frequency of each grid box to 75% of its base value across the whole study domain and keep the base spatiotemporal precipitation patterns over each specific region.

For convenience in identifying and describing all the sensitivity tests, we name them after their precipitation frequency and intensity scaling factors. For instance, the case f0.5i2 represents the simulation with half the base precipitation frequency and twice the base precipitation intensity, while the case f1i1 indicates the control simulation with a base frequency and intensity. We carry out more than 20 sensitivity model simulations to cover various precipitation intensities and frequencies as shown in Table 1.

The abundance of atmospheric aerosols is determined by both the aerosol emission rates and their atmospheric residence times, i.e., their lifetimes. The average atmospheric lifetimes of aerosols are calculated as

$$lifetime = \frac{burden}{removal\ rate} = \frac{burden}{dry\ deposition\ rate + wet\ deposition\ rate} \qquad (3)$$

Therefore, more efficient wet scavenging would lead to shorter atmospheric aerosol lifetimes.

We then examine the long-term changes in precipitation characteristics for various regions around the world in past decades. We first analyze changes in the precipitation between two 7-yr periods

(2008-2014 vs. 2001-2007) based on an observational dataset, the 3-Hour Realtime Tropical Rainfall

Measuring Multi-Satellite Precipitation Analysis version 7 (TRMM3B42v7, short for TRMM,

https://pmm.nasa.gov/TRMM). TRMM (3B42v7) performances better than the previous version of

satellite products (3B42v6), though there are still problems in detecting precipitation events with low

precipitation rates [*Maggioni et al*., 2016]. We then examine three reanalysis datasets with longer

temporal coverage (2001-2010 vs 1981-1990): the National Centers for Environmental Prediction

(NCEP) reanalysis dataset [*Kalnay et al.*, 1996], the NCEP-DOE AMIP-II (NCEP2) reanalysis dataset

[*Kanamitsu et al.*, 2002], and NASA's Modern-Era Retrospective analysis for Research and Applications

(MERRA) dataset [*Rienecker et al.*, 2011]. These datasets have different resolutions and spatial

coverage. TRMM only covers 60$^{\circ}$N-60$^{\circ}$S, while other datasets cover the whole globe. The resolutions

($^{\circ}$lon x $^{\circ}$lat x hour) for TRMM, NCEP, NCEP2, and MERRA are 0.25x0.25x3, 2.5x2.5x6, 2.5x2.5x6,

2.5x2x1, respectively. We regrid the TRMM dataset from 0.25x0.25 to 2.5x2.5 ($^{\circ}$lon x $^{\circ}$lat) to reduce the

computational cost and the relative errors at small precipitation rates [*Huffman et al.*, 2007; *Gehne et

al.*, 2016]. By combining the resulting sensitivities of BC lifetimes to precipitation characteristics with

the results of the long-term trends in precipitation characteristics, we then estimate the impacts of

long-term changes in precipitation characteristics on the atmospheric lifetime of BC.

**3 Results**

The global annual mean lifetime of BC is calculated at 5.29 days in our control simulation (Fig. 1). This

value is similar to the results of a previous study, which stated that the lifetime of BC would be around

one week [*Ramanathan and Carmichael*, 2008]. Our result also agrees with the lifetime of $5.8 \pm 1.8$ days

simulated by the GEOS-Chem model [*Park et al.*, 2005] and the 5.4 days result simulated by the

ECHAM5-HAM model [*Stier et al.*, 2005]. For 13 models in AeroCom, the lifetimes of BC from

anthropogenic fossil fuel and biofuel sources are simulated to be from 3.5 to 17.1 days, with 5.9 days as

the median value [*Samset et al.*, 2014].

We first compare the results of the control run with other simulations with the same precipitation

frequency (f1i0.25, f1i0.5, f1i1, f1i2, and f1i4) to examine the sensitivity of BC lifetime to precipitation

intensity (Fig. 1a). We find that an increase in precipitation intensity leads to decreases in both the BC

lifetime and the sensitivity of the BC lifetime to precipitation intensity. That is, the impact of precipitation intensity on BC aerosols is saturated when the intensity is very high, which is consistent with a previous study [*Fang et al.*, 2011]. We then compare the control run with other simulations with the same precipitation intensity (f0.1i1, f0.25i1, f0.5i1, f0.75i1, and f1i1) to study the sensitivities of the BC lifetime to precipitation frequency (Fig. 1b). Again, the BC lifetime responds non-linearly to the changes in precipitation frequency, and the sensitivity decreases with increases in precipitation frequency.

When we compare the simulations with a common precipitation amount (f0.1i10, f0.25i4, f0.5i2, f0.75i1.33, and f1i1), we find that the BC lifetime increases with increasing precipitation intensity (Fig. 1c). For example, case f0.1i10 has an annual average BC lifetime of 7.86 days, which is much longer than the 5.29 days of the control simulation (case f1i1). This indicates that the sensitivity of the BC lifetime to precipitation frequency is stronger than that to the precipitation intensity.

The calculated efficiency of wet scavenging can be affected by model parameterizations. We first examine the possible impacts on our results from the parameterization on the hygroscopicity of aerosols. With the default parameterization in GEOS-Chem, 20% of the fresh BC emissions are assumed to be hydrophilic. We set up sensitivity runs with another parameterization, where all BC are assumed to be hydrophilic. With these two different parameterization schemes, we examine the changes in the BC lifetime between two scenarios (f1i1 vs. f0.75i1.33) respectively. We find that with the default setting in GEOS-Chem, the atmospheric lifetime of BC under the f0.75i1.33 scenario is slightly higher than the f1i1 scenario by 0.4%. In comparison, if all the BC is assumed to be hydrophilic, the BC lifetime under the f0.75i1.33 scenario would be 3.6% higher. This implies that for hydrophilic aerosols, the sensitivity to precipitation frequency would be even higher.

We also evaluate the impacts on wet scavenging from aerosol size with sensitivity simulations. If we assume the aerosols to be in coarse mode, we find that it would lead to more efficient scavenging and consequently much shorter lifetime (compared to the default setting in GEOS-Chem that all BC aerosols are in accumulation mode). However, there are no significant effects on the relative sensitivities to precipitation frequency vs. intensity – the percentage change in BC lifetime between the f1i1 and f0.75i1.33 scenarios is very similar to the cases with parameterization for accumulation mode (0.3% vs.

0.4%). This indicates that the relative sensitivity of the BC lifetime to precipitation frequency and precipitation intensity is not significantly affected by the parameterization of particle size in the wet scavenging scheme in GEOS-Chem. It is worth noting that our model does not resolve the size of precipitation droplet, which can also affect the efficiency of wet scavenging.

The stronger sensitivity of the BC lifetime to precipitation frequency than that to intensity implies that an increase in the total precipitation amount does not necessarily lead to a decrease in the BC lifetime. This is better illustrated in Fig 2, which shows the BC lifetime as a function of the precipitation intensity and frequency based on 20 cases (f0.25, f0.5, f0.75, f1 versus i0.5, i1, i1.33, i2, i4). Compared with the control scenario (i.e., f1i1, the base precipitation intensity and frequency, as labeled by the black star), any point in the area between the two solid curves (the green one shows a constant total precipitation amount and the red one shows a constant BC lifetime) would have a higher total precipitation amount and a longer BC lifetime. This indicates that, even with an increased total precipitation, the BC lifetime (and hence the atmospheric concentrations of BC) can still increase if the precipitation frequency decreases significantly. This feature may help explain the decrease of the wet deposition flux found in wetter future climate simulations, despite their slightly increased total precipitation amounts [*Xu et al.*, 2018].

The lifetime contour plot in Fig. 2 can be employed as a simple tool to help us understand the impacts of long-term changes in precipitation on atmospheric aerosols, so we also investigate the long-term trends in the precipitation characteristics over the past decades for various regions around the world. In considering the spatial variations of precipitation patterns and their long-term trends, we divide the global continental regions into multiple subcontinental areas to better resolve the spatial variations (Fig. 3). We first carry out an analysis based on precipitation data from the TRMM dataset. The changes in the average precipitation intensities and frequencies between the periods of 2008-2014 and 2001-2007 for each region are shown as ratios in Fig. 4, with the width and height of the blocks indicting the standard errors of the calculated percentage changes in precipitation frequency and intensity, respectively. Although these TRMM data only cover 14 years, the standard errors as shown in Figure 4 indicate that the changes in precipitation intensity and frequency over most regions are statistically significant. We find that during these 14 years, the average precipitation intensity has increased over most regions, but the average precipitation frequency has decreased over more than one third of the total regions

including western North America (nwNA and swNA), southern South America (sSA), western Europe (wEU), southern Africa (sAF), and southwestern Asia (swAS). Based on the TRMM dataset, we find that almost all (5 out of 6) of the regions with decreasing precipitation frequency are expected to experience longer atmospheric aerosol lifetimes.

Since the TRMM data only cover a relatively short period, we make similar analyses with three reanalysis datasets (NCEP, NCEP2, and MERRA) to cover a longer time period (2001-2010 vs. 1981-1990) (Fig. 5). We find that, similar to the TRMM data, all the three reanalysis datasets show increasing trends for precipitation intensity over most regions but more divergent trends for precipitation frequency in the past decades. The NCEP data show that precipitation frequency has decreased over about two thirds of the total regions while NCEP2 and MERRA data show decreasing precipitation frequency over one third and half of the total regions, respectively. In addition, even when the different datasets indicate the same direction for the precipitation change over a specific region, the magnitude of the changes may vary significantly across datasets. For example, the derived changes in the average precipitation intensity over neNA (northeastern North America) based on NCEP, NCEP2, and MERRA data are +8%, +12%, and +3% respectively. These variations across different data sources reflect the significant uncertainties associated with these datasets, as reported earlier [e.g., *Trenberth and Christian*, 1998; *Trenberth et al.*, 2011; *Gehne et al.*, 2016].

On the other hand, previous analysis on global land-average precipitation showed that various reanalysis datasets have similar trends and interannual variability with other gauge- and satellite-based datasets during 2001-2010, though the estimated trend of precipitation varies based on temporal and spatial scales [*Gehne et al.*, 2016]. In addition, our study focuses on the changes over continental regions, where the precipitation data in the reanalysis datasets are found to be more reliable than over the ocean regions [*Trenberth et al.*, 2011]. Therefore despite the uncertainties associated with each meteorological dataset, we can use Fig. 5 to estimate the expected changes in the atmospheric BC lifetimes for certain regions, especially for those regions showing consistent trends across different datasets. Assuming the effects of precipitation on wet deposition is the only factor that affects the atmospheric BC aerosol lifetimes, all three datasets indicate that atmospheric BC aerosol lifetimes could have decreased in the northern regions of North America (neNA and nwNA), the northwestern and southern regions of South America

(nwSA and sSA), South Africa (sAF), and North Oceania (nOC). All three meteorological datasets show increasing trends in aerosol lifetimes over southwestern North America (swNA), Middle Africa (mAF), and South Oceania (sOC), which imply increasing trends for the concentrations of particulate matter (PM) over these regions, driven by changes in precipitation. At the regional scale, precipitation changes over the past 30 years can easily lead to perturbations in atmospheric BC lifetimes by 10% or higher.

We should note that there are some caveats for our idealized sensitivity simulations. The way we reduce precipitation frequency in the model (based on a stochastic function as discussed in section 2) can be very different from climate-driven precipitation change in the real world. The globally uniform scaling factors applied to precipitation intensity do not account for the spatial variations. As a consequence, the sensitivities of BC lifetime to precipitation changes over a specific region may be

different from those shown in Fig. 2. To partly address this issue, we have constructed some regional contour plots similar to that in Fig. 2 but based on sensitivities of BC lifetime for those specific regions (Fig. 6). Comparison of these regional contours with the global one indicate some differences in the sensitivity of BC to precipitation changes, but generally less than 3%. In addition, to clearly demonstrate that the BC lifetime has different sensitivities to precipitation intensity and frequency, our

sensitivity simulations cover a wide range of precipitation intensity and frequency. Some of these applied perturbations are significantly larger than those induced by climate change, especially at large (such as regional or global) scales. Therefore, simple interpolation of some results from this study in examining the effects from climate change may introduce some uncertainties. Our results are also affected by the limitations from the meteorology datasets. Although the TRMM and the reanalysis

datasets used in this study represent some of the best meteorological datasets available, each of them has their own shortcomings - the observational datasets are more reliable, but only cover a relatively short time period of 14 years; the reanalysis datasets cover longer periods, but are less reliable due to known issues such as the bias in moisture budget [e.g., *Trenberth and Christian*, 1998; *Trenberth et al.*, 2011; *Gehne et al.*, 2016].

**4 Conclusions and Discussion**

The efficiency of the wet scavenging of atmospheric aerosols is affected by not only the precipitation amount but also the precipitation patterns. Our results, based on sensitivity simulations with the GEOS-Chem model, show that the atmospheric lifetimes of BC are more sensitive to precipitation frequency than precipitation intensity, and as a consequence, increases in the total precipitation amount do not always lead to a more efficient wet scavenging of atmospheric aerosols. The sensitivities of the atmospheric lifetimes of aerosols to the precipitation characteristics derived from our model simulations offer a simple and convenient tool for us to better examine the implications of long-term changes in precipitation (including the total amounts and patterns) for atmospheric aerosols in various regions.

Analysis of satellite data (TRMM) for the past 14 years (2001-2014) reveal that precipitation intensity has increased in most regions. On the other hand, decreasing precipitation frequency are found in some regions such as western North America, southern South America, western Europe, southern Africa, and southwestern Asia. The decreases in precipitation frequency could lead to increases in atmospheric aerosol lifetimes over these regions. Our further analyses based on three meteorological datasets (NCEP, NCEP2, and MERRA) for the past decades (1981-2010) show increases in precipitation intensities over most continental regions, but significant decreases in precipitation frequency are identified over some regions. These changes in precipitation characteristics affect the wet deposition of aerosols and consequently the total burdens of aerosols and their atmospheric lifetimes. Despite the significant uncertainties associated with meteorological data, we find that the changes in precipitation intensity and frequency over the past 30 years could have led to perturbations in the regional atmospheric aerosol lifetimes by 10% or higher. Our results are consistent with *Kloster et al.* [2010] and *Fang et al.* [2011] who reported increasing atmospheric aerosol burden due to climate change, although their results are based on future climate change. We also find that all three meteorological databases are consistent to show that the changes in precipitation intensity and frequency over the past decades have led to decreases in atmospheric aerosol lifetimes over the northern regions of North America, northwestern and southern regions of South America, South Africa, and North Oceania. They are also consistent in indicating increasing trends of atmospheric aerosol lifetimes in the southwestern region of North America, Middle Africa, and South Oceania. The increasing trends in atmospheric aerosol lifetimes over these regions

driven by the changes in precipitation intensity and frequency in the context of global climate change could pose challenges for the local PM air qualities. It should be noted that the results from this work can be affected by the parameterization in the GEOS-Chem model and have certain limitations. Our study does not account for the impacts of precipitation on wildfires which can emit massive amount of aerosols including BC [*Dawson et al.*, 2014].

**Acknowledgements**

We thank all those who have contributed to the datasets we used in this study. The NCEP reanalysis data and NCEP-DOE AMIP-II reanalysis data were provided by NOAA/OAR/ESRL PSD, Boulder, Colorado, USA, and were accessed through their web site at http://www.esrl.noaa.gov/psd/. The MERRA data used in this study were provided by the Global Modeling and Assimilation Office (GMAO) at the NASA Goddard Space Flight Center through the NASA GES DISC online archive. We thank Dr. Hongyu Liu and Dr. Bo Zhang for fruitful discussion. Superior, a high performance computing cluster at Michigan Technological University, was used to obtain the results presented in this publication. S. Wu acknowledges sabbatical fellowship from the Ocean University in China. This publication was made possible by a U.S. EPA grant (grant 83518901). Its contents are solely the responsibility of the grantee and do not necessarily represent the official views of the U.S. EPA. Further, the U.S. EPA does not endorse the purchase of any commercial products or services mentioned in this publication.

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

**Table 1. Series of sensitivity model simulations carried out in this study.**

| Model simulations | Objective | Case names |
|---|---|---|
| Constant precipitation frequency (Fig. 1a) | To study the sensitivity of BC lifetime to precipitation intensity | f1i0.25, f1i0.5, f1i1, f1i2, and f1i4 |
| Constant precipitation intensity (Fig. 1b) | To study the sensitivity of BC lifetime to precipitation frequency | f0.1i1, f0.25i1, f0.5i1, f0.75i1, and f1i1 |
| Constant precipitation amount (Fig. 1c) | To compare the sensitivity of BC lifetime to precipitation intensity and precipitation frequency | f0.1i10, f0.25i4, f0.5i2, f0.75i1.33, and f1i1 |
| Hygroscopicity of aerosols (100% vs 20% BC in fresh | To examine the impacts on wet deposition from the | f1i1 and f0.75i1.33 |

| emissions are assumed to be hydrophilic) | parameterization on the hygroscopicity of aerosols | |
|---|---|---|
| Aerosol size (BC aerosols are assumed to be in coarse mode vs accumulation mode) | To examine the impacts on wet scavenging from the parameterization on the size of aerosols | f1i1 and f0.75i1.33 |
| Contour of BC lifetime (Fig. 2, 4-6) | To plot BC lifetime as a function of the precipitation intensity and frequency | f0.25i0.5, f0.25i1, f0.25i1.33, f0.25i2, f0.25i4, f0.5i0.5, f0.5i1, f0.5i1.33, f0.5i2, f0.5i4, f0.75i0.5, f0.75i1, f0.75i1.33, f0.75i2, f0.75i4, f1i0.5, f1i1, f1i1.33, f1i2, and f1i4 |

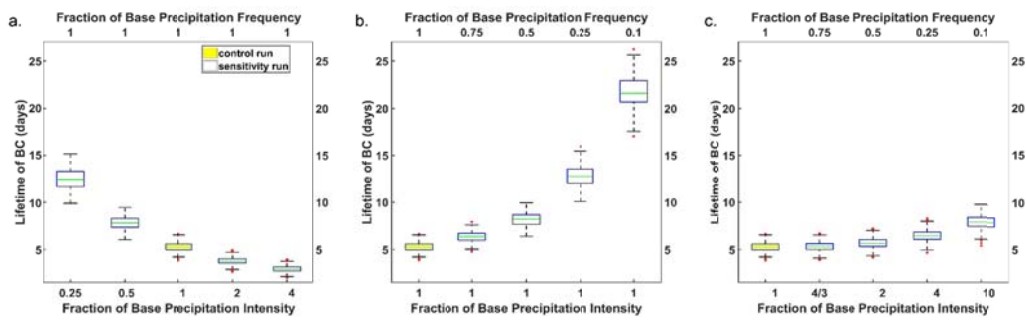

**Figure 1: Impacts of the precipitation characteristics on the atmospheric lifetime of BC under given a) constant precipitation frequency; b) constant precipitation intensity; and c) constant precipitation amount. The top x-axis reflects the precipitation frequency set in each perturbation test, shown as fractions of base precipitation frequency. Base precipitation frequency is the precipitation frequency used in the control case. Similarly, the bottom x-axis reflects the settings of precipitation intensity in the perturbation tests. The box plot shows the probability distribution of BC lifetime for each case, where the top and bottom edges of each box show the third and first quartiles, respectively; the green central bar shows the median; the whisker shows the range of the non-outliers that cover 99.3% of the data, assuming normally distributed data; and the red plus shows the outliers.**

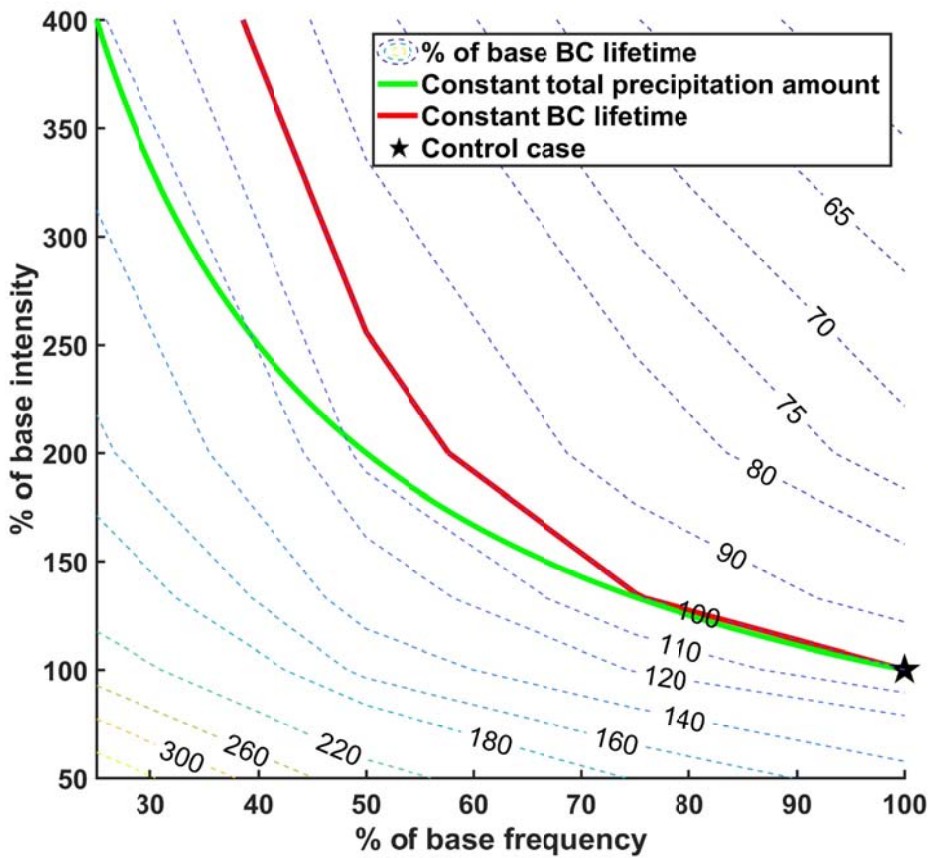

**Figure 2: Model calculated BC atmospheric lifetime as a function of precipitation intensity and frequency. The dashed contour lines indicate the atmospheric lifetimes of the black carbon aerosols from the interpolation of 20 cases, which show the potential changes of BC lifetimes from the base BC lifetime (in the control run) driven by the changes of precipitation intensity and frequency. The green solid line represents a**
395 **total precipitation equal to that of the base simulation (control run). The red solid line indicates the conditions leading to atmospheric black carbon aerosol lifetimes that match the base simulation (control run).**

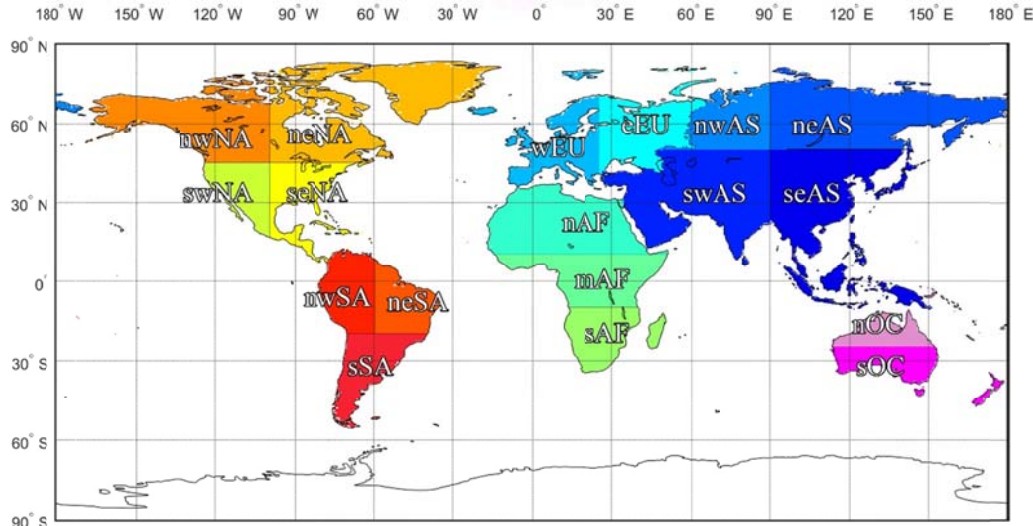

**Figure 3: The definitions of the continental regions in this study. The uppercase letters in the region names**
**represent the names of their continents: North America (NA), South America (SA), Europe (EU), Africa**
**(AF), Asia (AS), and Oceania (OC). The lowercase letters in the region names represent the subregions inside**
**the continent: north (n), south (s), west (w), east (e), middle (m), northwest (nw), northeast (ne), southwest**
**(sw), and southeast (se).**

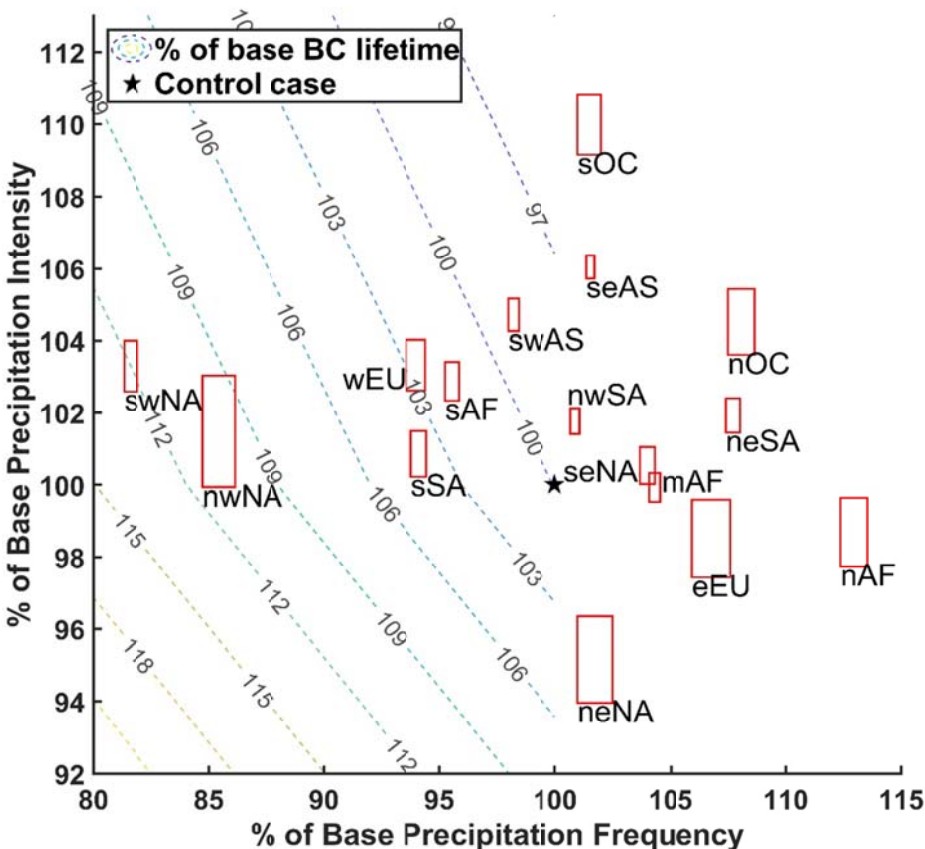

**Figure 4: The potential change of atmospheric BC aerosol lifetime driven by the changes between the two periods (2008-2014 and 2001-2007) in precipitation characteristics based on meteorological datasets TRMM. The dashed contours are the same as in Fig. 2, which indicate the atmospheric lifetimes of the black carbon aerosols from the interpolation of 20 cases and show the potential changes of BC lifetimes from the base BC lifetime (in the control run) driven by the changes of precipitation intensity and frequency. Red blocks show the changes of precipitation intensities and frequencies, with the size of the block showing the standard error of the percentage changes.**

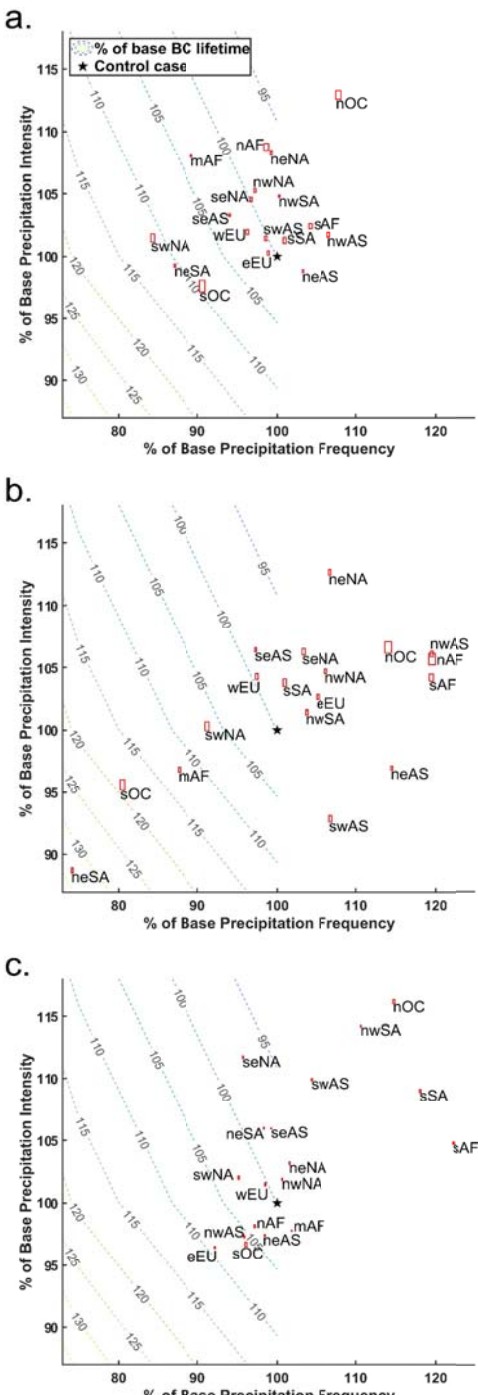

**Figure 5: The potential change of atmospheric BC aerosol lifetime driven by the changes between the two periods (2001-2010 and 1981-1990) in precipitation characteristics based on multiple meteorological datasets: a). NCEP; b). NCEP2; c). MERRA. The dashed contours are the same as in Fig. 2, which indicate the atmospheric lifetimes of the black carbon aerosols from the interpolation of 20 cases and show the potential changes of BC lifetimes from the base BC lifetime (in the control run) driven by the changes of precipitation intensity and frequency. Red blocks show the changes of precipitation intensities and frequencies, with the size of the block showing the standard error of the percentage changes.**

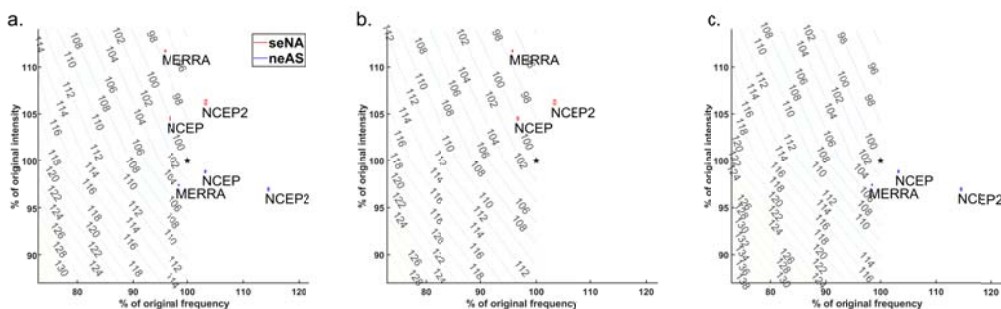

**Figure 6: Compare the contours calculated on the global and regional scale: a). global; b). southeast North America (seNA); c). northeast Asia (neAS). The contours indicate the atmospheric lifetimes of the black carbon aerosols from the interpolation of 20 cases and show the potential changes of BC lifetimes from the base BC lifetime (in the control run) driven by the changes of precipitation intensity and frequency. The contour calculated on the global scale is the same with Fig. 2. seNA and neAS are two most extreme cases among all regions, with the smallest and largest sensitivities between BC lifetimes and precipitation changes.**
