# Peer review of "Sensitivity of atmospheric aerosol scavenging to precipitation intensity and frequency in the context of global climate change"

_Atmospheric Chemistry and Physics, 2017_

## Referee Comment (RC1) · Anonymous Referee #3 · 8 Jan 2018

Hou et al. systematically investigate the effect of precipitation frequency and intensity on aerosol scavenging using a coarse resolution global model with a rather simplistic description of aerosol scavenging. The topic is especially interesting since the changes in precipitation characteristics (e.g. more extreme precipitation and possibly less drizzle) constitute an important contributor to the climate change signal. While the finding that the change of the black carbon lifetime in a changing climate might be dominated by changes in precipitation frequency and not in precipitation amount does not seem overly surprising in the light of the cited literature, this study nevertheless seems very interesting and useful to me, especially since to my knowledge this study represents the first attempt to investigate the topic in such a focused and systematic fashion. The

[Discussion paper]

[Figure]

study nicely explains why an increase in total precipitation amount does not necessarily lead to a decrease in aerosol lifetime (independent of changes in spatial patterns that may for example impact some regions with high emissions more than others). In my opinion the manuscript serves to highlight a rather interesting and important topic and in spite of some limitations it can serve as a very good base for further studies. I recommend to publish the manuscript subject to minor revisions.

Specific comments/suggestions

1. l. 86ff: I think that (a) scaling the precipitation by a uniform factor for each grid box and (b) using a stochastic function where precipitation is turned off regardless of whether it is heavy or light precipitation may in principle lead to different outcomes compared to what might be expected from climate change (in which e.g. strong precipitation intensity may be enhanced while weak precipitation may decrease or remain unchanged and models also suggest very distinct spatial patterns) and I am not sure that the results from these very idealized sensitivity tests can be used to deduce a quantitatively correct answer for the climate change signal. I suggest to discuss this point. Also, as far as possible, I would appreciate if the authors could put their estimate of the change in aerosol lifetime into the context of other estimates from the literature, e.g. in the conclusion section in l. 241, although most of the existing literature estimates will not be directly comparable since they look at different regions and times. For example, Fang et al. (2011) estimate a change in lifetime for their SAt tracer. I also wonder if it would make sense to construct Fig. 2 for each region separately?

2. Especially the time period covered by the TRMM dataset is rather short, so that influences of internal variability are likely to play some role at least on a regional bases. On the other hand, the increase in precipitation intensity is consistent with expectations in a warming climate. It would also be interesting to see what part of the changes in precipitation frequency in Fig. 4 may be associated with internal variability, although I realize that this is outside the scope of this study. I think it would nevertheless be good to more explicitly mention that some of the regional trends may at least in part

be due to internal variability e.g. in line 221. For example CMIP5 model simulations suggest that the effect of internal variability even on multi-decadal regional precipitation trends can be rather large, especially for small regions. The global average changes, on the other hand, are much more directly related to the forcing strength. The large spread in the values of precipitation frequency in Fig. 4 may also be an indication of internal variability, although I am not sure if one can obtain an estimate based on the existing literature. Further research which is outside the scope of this work may be required to quantify this. One way to "filter out" the effect of internal variability might be to compute the average change in the BC lifetime over all regions, although one could argue that this also means loosing other information that is contained in the regional averages (e.g. differences due to different characteristics of the regions) and that the regional lifetimes are generally of larger interest than the global average. My recommendation would nevertheless be to compute the 30-year changes of the global average BC lifetime for all the land areas (with the contributions from the individual regions weighted by the size of the individual regions) and also for the entire globe and to state the values in the conclusion section. This may then also facilitate a more meaningful discussion of the results from this study in relation to existing literature.

3. The parameter range that is explored in Fig. 2 seems very large in the context of global climate change and there seem to be relatively few sensitivity simulations that are in the range of expected climate change. On the other hand, any potential bias that results from this will most likely not be overly large in the light of other uncertainties that stem from incomplete knowledge of actual and expected precipitation changes, uncertainties in the scavenging formulation, and possibly also uncertainties related to the design of the study (see my point one #1 above).

Other specific comments/suggestions/questions:

1. In the introduction, there are a few cases (Salzmann et al., line 28; Trenberth et al., 2007, line 29; Trenberth et al., 2011, line 31; Dawson et al., 2007, line 36; Fang et al., 2011, line 40) in which it might be nice to know what the cited findings are based

on (e.g. observations, regional/global model, theoretical arguments, combination of modeling and observations, models constrained by observations, etc).

2. l. 59: in addition to the URL, please also cite at least one paper that describes GEOS-Chem, even if it not exactly the version that is used here.

3. l. 72: unit of P?

4. l. 78: did the authors check whether the results are sensitive to this definition?

5. l. 115f: did the authors check whether the result is sensitive to this?

6. l. 157 and lines 165ff: good points that are nicely explained.

7. l. 178: are those the standard deviations of the yearly means?

8. l. 240 ff: "precipitation changes" is used here and also further below. It would be better to be more specific regarding whether this is mostly frequency or intensity.

9. l. 251: "feedbacks" are usually understood to be mediated by sea surface temperature (SST) change. In a model run in which SSTs are prescribed based on observations, the effect of aerosol on SST during this period is actually taken into account. But the authors are right in the sense that assessing the magnitude of the feedbacks is not possible in such a setup.

Suggestions for technical corrections

l. 15: omit "simulation" l. 19: aerosols -> aerosol l. 26: other atmospheric elements -> soluble trace gases l. 67: details -> detail l. 86: control -> the control l. 93: simulation tests -> sensitivity tests l. 98: rate -> rates l. 104: precipitations -> precipitation l. 108: We -> . We l. 126 control -> the control l. 149: that -> that this l. 200: same -> the same l. 232: have -> has l. 315: year? l. 346: control -> the control

Fig 1: please increase the size of the labels (and/or magnify the figure) and increase the resolution so that the figure can be magnified on the screen. Please also consider

increasing the resolution of Fig. 5.

---

## Referee Comment (RC2) · Anonymous Referee #2 · 24 Jan 2018

This is a potentially interesting paper on changes in wet deposition due to precipitation intensity and amount changes. However, there are several issues that need to be addressed before the paper can be accepted: 1. The methods is not complete, and more details on the experiments and a better descirption of the cases (in a table) need to completed. More details below. 2. The trend in the precipitation using either the TRIMM or the reanalyses is unlikely to be robust or good enough for this analysis. The Renalysis are well know to have trouble with the moisture budget, while the TRIMM time series is too short.

I wonder if the authors don't want to just do a correlation between the annual average

precipitation in different regions and the wet deposition lifetime in that region, and see if there isn't a robust signal in that. Then you can still make some statements about how different regions are likely to move, based on climate projections or longer term precipitation trends. I would bet, if your results are robust, that you will get a good relationship just with annual averages (or seasonal), and then you can more safely extrapolate into the future.

More details:

"Our study, based on the GEOS-Chem model simulation, shows that the removal efficiency and hence the atmospheric lifetime of aerosols have significantly higher sensitivities to precipitation frequencies than to precipitation intensities, indicating that the same amount of precipitation may lead to different removal efficiencies of atmospheric aerosols." Please make it clear that this is dependent on the way that you have included wet deposition, but that we don't really know the right answer.

"We first analyze changes in the precipitations between two 7-yr periods (2008-2014 vs. 2001-2007)" This is a very short time scale to talk about: is this going to be interpretable? Please go into the details of statistical significance and interannual variability and what your goal is with such short time scale differences.

61:"model does not simulate meteorology;": it does simulate meteorology, but it does not do this prognostically, it just forces it from data.

67: Change:"in details by" to in detail by

68: "The efficiency of wet scavenging is very sensitive to the hydrophilicity of BC." Please make clear that this is a result from your study.

96: f0.5i2: I appreciate that you tried to make your case names make sense, but they are still unintelligible, so you probably want a table describing all your cases, and try to NOT use your case name, but rather use English whenever possible.

"Our result also agrees with the lifetime of 5.8 $\pm$ 1.8 days simulated by the GEOS-

Chem model [Park et al., 2005] and the 5.4 days result simulated by the ECHAM5-HAM model [Stier et al., 2005]." How does it compare to other models in AEROCOM? Why just compare to two previous studies?

140:" The efficiency of wet scavenging can be affected by model parameterization. We first examine the impacts on our results from the parameterization on the hygroscopicity of aerosols. We compare the changes in the BC lifetime between two scenarios (f1i1 vs. f0.75i1.33) with alternative parameterization schemes." How did you change the hygroscopicity? Please add to the methods section.

150: "We also evaluate the impacts on wet scavenging from aerosol size with sensitivity simulations. If we assume the aerosols to be in coarse mode, we find that would lead to more efficient scavenging and 150 consequently much shorter lifetime (compared to the default setting in GEOS-Chem that all BC aerosols are in accumulation mode)." Plesae describe your fine and coarse mode depencies in the model so that we understand why this occurred. Overall in the methods you need to repeat a full description of your wet deposition algorithm, as your results could be completely sensitive to how you have parameterized this.

"We find that during these 14 years, the average precipitation intensity has increased over most regions, but the average precipitation 180 frequency has decreased over more than one third of the total regions including western North America (nwNA and swNA), southern South America (sSA), western Europe (wEU), southern Africa (sAF), and southwestern Asia (swAS)." This is a very short time period to argue for increases or decreases: this could just be interannual variability. Do you really want to argue for increases or decrsease? If so, show a statistically significant difference, etc. I would argue a better way to do it, is just use annual averages, which will allow you to use more data (as described above).

"By combining these precipitation changes for various regions as shown in Fig. 4 with the relationship between precipitation characteristic and the BC lifetime as illustrated

[Figure]
* * *
Interactive
comment

in Fig. 2, we can analyze the long-term changes in the atmospheric aerosol lifetimes driven by precipitation changes." Because I don't believe you have a long enough time series, I don't believe you can extend it, unfortunately. Please think about doing this analysis in a much more robust manner (with error bars showing the trends are important enough, believeable, etc) or just pull this section out of the paper.

"Since the TRMM data only cover a relatively short period, we make similar analyses with three reanalysis datasets (NCEP, NCEP2, and MERRA) to cover a longer time period (2001-2010 vs. 1981-1990) (Fig. 5). We find that, similar to the TRMM data, all the 195 three reanalysis datasets show increasing trends for precipitation intensity over most regions but more divergent trends for precipitation frequency in the past decades." Here you might have enough data to talk about this, but still very short time period. Again, show the standard deviations, show that they are signficiant, suppor tiwth other studies that try to show trends across such very short time periods in such a highly variable value (precipitation).

Also, there are significant problems with the moisture budgets in the reanalyses: are you sure you even want to do this? Might be better to just use climate model output because of the problems with inconsistencies in the data (please see all the papers by Kevin Trenberth showing the very large warts in the moisture budgets for all the reanalyses; not just one paper).

---

## Referee Comment (RC3) · Anonymous Referee #2 · 26 Jan 2018

Please notice that the argument that the frequency of precipitation is more useful than intensity for understanding wet deposition lifetime changes in context of changes in aerosol lifetime for the LGM/current was used in the Mahowald et al., 2011 paper in Quaternary Science Reviews, which might help your arguments here.
* * *

---

## Author Comment (AC1) · 21 Mar 2018

**Point-to-Point Response for "Sensitivity of atmospheric aerosols to precipitation characteristics"**

We thank both referees for their very helpful comments. We have carried out further analyses and also revised the manuscript following the referees' comments. We have made itemized responses to all the comments as described below. The referees' comments are repeated below in the blue and italicized text and our responses are in normal font.

**Response to Referee #2**

*This is a potentially interesting paper on changes in wet deposition due to precipitation intensity and amount changes. However, there are several issues that need to be addressed before the paper can be accepted: 1. The methods is not complete, and more details on the experiments and a better descirption of the cases (in a table) need to completed. More details below.*

Thank you for pointing this out. We have added a new table (Table 1) in the MS to summarize and better describe the various cases done in this study. We have also provided more clarification throughout the MS as detailed below in response to specific comments and questions.

**Table 1. Series of sensitivity model simulations carried out in this study.**

| Model simulations | Objective | Case names |
|---|---|---|
| Constant precipitation frequency (Fig. 1a) | To study the sensitivity of BC lifetime to precipitation intensity | f1i0.25, f1i0.5, f1i1, f1i2, and f1i4 |
| Constant precipitation intensity (Fig. 1b) | To study the sensitivity of BC lifetime to precipitation frequency | f0.1i1, f0.25i1, f0.5i1, f0.75i1, and f1i1 |
| Constant precipitation amount (Fig. 1c) | To compare the sensitivity of BC lifetime to precipitation intensity and precipitation frequency | f0.1i10, f0.25i4, f0.5i2, f0.75i1.33, and f1i1 |
| Hygroscopicity of aerosols (100% vs 20% BC in fresh emissions are assumed to be hydrophilic) | To examine the impacts on wet deposition from the parameterization on the hygroscopicity of aerosols | f1i1 and f0.75i1.33 |

| | | |
|---|---|---|
| Aerosol size (BC aerosols are assumed to be in coarse mode vs accumulation mode) | To examine the impacts on wet scavenging from the parameterization on the size of aerosols | f1i1 and f0.75i1.33 |
| Contour of BC lifetime (Fig. 2, 4-6) | To plot BC lifetime as a function of the precipitation intensity and frequency | f0.25i0.5, f0.25i1, f0.25i1.33, f0.25i2, f0.25i4, f0.5i0.5, f0.5i1, f0.5i1.33, f0.5i2, f0.5i4, f0.75i0.5, f0.75i1, f0.75i1.33, f0.75i2, f0.75i4, f1i0.5, f1i1, f1i1.33, f1i2, and f1i4 |

*2. The trend in the precipitation using either the TRIMM or the reanalyses is unlikely to be robust or good enough for this analysis. The Renalysis are well know to have trouble with the moisture budget, while the TRIMM time series is too short.*

Indeed, although the TRMM and the reanalysis datasets used in this study represent some of the best meteorological datasets available, each of them has their own shortcomings - the observational datasets are more reliable, but only cover shorter time periods; while the reanalysis datasets cover longer periods, but are less reliable. That's why we decided to combine multiple datasets in this study, including TRMM, NCEP, NCEP2, and MERRA, to give us a better idea about the potential trends of precipitation characteristics. We have provided discussions about the choices of datasets, like –

"TRMM (3B42v7) performances better than the previous version of satellite products (3B42v6), though there are still problems in detecting precipitation events with low precipitation rates [*Maggioni et al.*, 2016]. "

"We regrid the TRMM dataset from 0.25x0.25 to 2.5x2.5 (°lon x °lat) to reduce the computational cost and the relative errors at small precipitation rates [*Huffman et al.*, 2007; Gehne *et al.*, 2016]."

"Since the TRMM data only cover a relatively short period, we make similar analyses with three reanalysis datasets (NCEP, NCEP2, and MERRA) to cover a longer time period (2001-2010 vs. 1981-1990) (Fig. 5)."

"These variations across different data sources reflect the significant uncertainties associated with these datasets, as reported earlier [e.g., *Trenberth and Christian*, 1998; *Trenberth et al.*, 2011; Gehne *et al.*, 2016]."

"Our results are also affected by the limitations from the meteorology datasets. Although the TRMM and the reanalysis datasets used in this study represent some of the best meteorological datasets available, each of them has their own shortcomings - the

observational datasets are more reliable, but only cover a relatively short time period of 14 years; the reanalysis datasets cover longer periods, but are less reliable due to known issues such as the bias in moisture budget [e.g., *Trenberth and Christian*, 1998; *Trenberth et al.*, 2011; *Gehne et al.*, 2016]."

*I wonder if the authors don't want to just do a correlation between the annual average precipitation in different regions and the wet deposition lifetime in that region, and see if there isn't a robust signal in that. Then you can still make some statements about how different regions are likely to move, based on climate projections or longer term precipitation trends. I would bet, if your results are robust, that you will get a good relationship just with annual averages (or seasonal), and then you can more safely extrapolate into the future.*

We presume by "annual average precipitation" the review refers to the annual total precipitation amount (i.e. average intensity x frequency). Then indeed, we believe that we would see a good correlation between the annual average precipitation and BC lifetime. However, a major point of our study here is to show that it's not just the annual average precipitation but also the patterns (say frequent drizzles vs occasional heavy rain events) would matter for BC scavenging. That is, due to the different sensitivities associated with precipitation intensity and frequency, the same annual average precipitation may lead to very different scavenging efficiency and consequently BC lifetime.

*More details:*

*"Our study, based on the GEOS-Chem model simulation, shows that the removal efficiency and hence the atmospheric lifetime of aerosols have significantly higher sensitivities to precipitation frequencies than to precipitation intensities, indicating that the same amount of precipitation may lead to different removal efficiencies of atmospheric aerosols." Please make it clear that this is dependent on the way that you have included wet deposition, but that we don't really know the right answer.*

We have modified the first part to "Our sensitivity model simulations, through some simplified perturbations to precipitation in the GEOS-Chem model, show that …"

*"We first analyze changes in the precipitations between two 7-yr periods (2008-2014 vs. 2001-2007)" This is a very short time scale to talk about: is this going to be interpretable? Please go into the details of statistical significance and interannual variability and what your goal is with such short time scale differences.*

We have added clarification in the MS –

"The changes in the average precipitation intensities and frequencies between the periods of 2008-2014 and 2001-2007 for each region are shown as ratios in Fig. 4, with the width and height of the blocks indicting the standard errors of the calculated percentage changes in precipitation frequency and intensity, respectively. Although these TRMM data only cover 14 years, the standard errors as shown in Figure 4 indicate that the changes in precipitation intensity and frequency over most regions are statistically significant."

*61:"model does not simulate meteorology;": it does simulate meteorology, but it does not do this prognostically, it just forces it from data.*

We have modified the text from "model does not simulate meteorology" to "model does not simulate meteorology prognostically".

*67: Change:"in details by" to in detail by*

The text in MS has been modified from "in details by" to "in detail by".

*68: "The efficiency of wet scavenging is very sensitive to the hydrophilicity of BC." Please make clear that this is a result from your study.*

We have clarified this part to - "In GEOS-Chem simulation, the BC aerosols are classified into two types based on their hygroscopicity (hydrophobic vs hydrophilic) and wet scavenging is more efficient for hydrophilic BC."

*96: f0.5i2: I appreciate that you tried to make your case names make sense, but they are still unintelligible, so you probably want a table describing all your cases, and try to NOT use your case name, but rather use English whenever possible.*

We have added a new table (Table 1) in the MS to describe all the perturbation tests(also shown above). We feel the concise name such as f0.1i1 works well when citing the specific case; the long English case name such as "a case with 0.1 times of base precipitation frequency and base precipitation intensity" appears too wordy and can easily break the reading flow. We hope the descriptions in table would help making the short names clearer.

*"Our result also agrees with the lifetime of 5.8 ± 1.8 days simulated by the GEOS Chem*

*model [Park et al., 2005] and the 5.4 days result simulated by the ECHAM5-HAM model [Stier et al., 2005]." How does it compare to other models in AEROCOM? Why just compare to two previous studies?*

We have compared to more models in AEROCOM. -- "For 13 models in AeroCom, the lifetimes of BC from anthropogenic fossil fuel and biofuel sources are simulated to be from 3.5 to 17.1 days, with 5.9 days as the median value [*Samset et al.*, 2014]."

*140:" The efficiency of wet scavenging can be affected by model parameterization. We first examine the impacts on our results from the parameterization on the hygroscopicity of aerosols. We compare the changes in the BC lifetime between two scenarios (f1i1 vs. f0.75i1.33) with alternative parameterization schemes." How did you change the hygroscopicity? Please add to the methods section.*

We have added more description about the sensitivity test on hygroscopicity in the text –

"We first examine the possible impacts on our results from the parameterization on the hygroscopicity of aerosols. With the default parameterization in GEOS-Chem, 20% of the fresh BC emissions are assumed to be hydrophilic. We set up sensitivity runs with another parameterization, where all BC are assumed to be hydrophilic. With these two different parameterization schemes, we examine the changes in the BC lifetime between two scenarios (f1i1 vs. f0.75i1.33) respectively. "

We have also included brief description on this in Table 1, showing that one case with "20% BC in fresh emissions are assumed to be hydrophilic (default)" and the other case with "100% BC in fresh emissions are assumed to be hydrophilic".

*150: "We also evaluate the impacts on wet scavenging from aerosol size with sensitivity simulations. If we assume the aerosols to be in coarse mode, we find that would lead to more efficient scavenging and consequently much shorter lifetime (compared to the default setting in GEOS-Chem that all BC aerosols are in accumulation mode)." Plesae describe your fine and coarse mode depencies in the model so that we understand why this occurred. Overall in the methods you need to repeat a full description of your wet deposition algorithm, as your results could be completely sensitive to how you have parameterized this.*

We have described the wet deposition algorithm for both accumulate and coarse mode: "The washout rate constant ($k$) is affected by the particle size and the form of precipitation. For washout by rain with precipitation rate $P$ (mm/h), $k = 1.1 \times 10^{-3}P^{0.61}$ for accumulation mode (aerosols with diameters between 0.04 µm and 2.5 µm) and $k = 0.92P^{0.79}$ for coarse mode (aerosols with diameter between 2.5 µm to 16 µm); for washout by snow with precipitation rate $P$, $k = 2.8 \times 10^{-2}P^{0.96}$ for accumulation

mode and $k = 1.57P^{0.96}$ for coarse mode [*Feng*, 2007, 2009]. The coefficients for accumulation-mode are used in calculating k for fine particles including BC in GEOS-Chem. "

*"We find that during these 14 years, the average precipitation intensity has increased over most regions, but the average precipitation 180 frequency has decreased over more than one third of the total regions including western North America (nwNA and swNA), southern South America (sSA), western Europe (wEU), southern Africa (sAF), and southwestern Asia (swAS)." This is a very short time period to argue for increases or decreases: this could just be interannual variability. Do you really want to argue for increases or decrsease? If so, show a statistically significant difference, etc. I would argue a better way to do it, is just use annual averages, which will allow you to use more data (as described above).*

We have updated this part to "The changes in the average precipitation intensities and frequencies between the periods of 2008-2014 and 2001-2007 for each region are shown as ratios in Fig. 4, with the width and height of the blocks indicting the standard errors of the calculated percentage changes in precipitation frequency and intensity, respectively. Although these TRMM data only cover 14 years, the standard errors as shown in Figure 4 indicate that the changes in precipitation intensity and frequency over most regions are statistically significant."

Our results indicate that though the results based on 14 years of data from TRMM (Fig. 4) show a much larger standard error than the results based on 30 years of data from other datasets (Fig. 5), there are still significant tendencies of increase and decrease in the results of 14 years for most regions.

*"By combining these precipitation changes for various regions as shown in Fig. 4 with the relationship between precipitation characteristic and the BC lifetime as illustrated in Fig. 2, we can analyze the long-term changes in the atmospheric aerosol lifetimes driven by precipitation changes." Because I don't believe you have a long enough time series, I don't believe you can extend it, unfortunately. Please think about doing this analysis in a much more robust manner (with error bars showing the trends are important enough, believeable, etc) or just pull this section out of the paper.*

We used the size of red block in Fig. 4 and Fig. 5 as the error bar to show the standard error. To clarify this point, we have added more discussion in MS – "The changes in the average precipitation intensities and frequencies between the periods of 2008-2014 and 2001-2007 for each region are shown as ratios in Fig. 4, with the width and height of the blocks indicting the standard errors of the calculated percentage changes in precipitation frequency and intensity, respectively. Although these TRMM data only cover 14 years,

the standard errors as shown in Figure 4 indicate that the changes in precipitation intensity and frequency over most regions are statistically significant."

*"Since the TRMM data only cover a relatively short period, we make similar analyses with three reanalysis datasets (NCEP, NCEP2, and MERRA) to cover a longer time period (2001-2010 vs. 1981-1990) (Fig. 5). We find that, similar to the TRMM data, all the 195 three reanalysis datasets show increasing trends for precipitation intensity over most regions but more divergent trends for precipitation frequency in the past decades." Here you might have enough data to talk about this, but still very short time period. Again, show the standard deviations, show that they are signficiant, suppor tiwth other studies that try to show trends across such very short time periods in such a highly variable value (precipitation).*

We used the size of red block in Fig. 4 and Fig. 5 as the error bar to show the standard error.

*Also, there are significant problems with the moisture budgets in the reanalyses: are you sure you even want to do this? Might be better to just use climate model output because of the problems with inconsistencies in the data (please see all the papers by Kevin Trenberth showing the very large warts in the moisture budgets for all the reanalyses; not just one paper).*

Good point. We have added more discussion in MS to acknowledge the limitation of the reanalysis datasets.–

"Our results are also affected by the limitations from the meteorology datasets. Although the TRMM and the reanalysis datasets used in this study represent some of the best meteorological datasets available, each of them has their own shortcomings - the observational datasets are more reliable, but only cover a relatively short time period of 14 years; the reanalysis datasets cover longer periods, but are less reliable due to known issues such as the bias in moisture budget [e.g., *Trenberth and Christian*, 1998; *Trenberth et al.*, 2011; *Gehne et al.*, 2016]."

By combining multiple datasets including TRMM and the reanalyses data, we are hoping that the analyses can still offer us some insights on some likely trends in precipitation.

*Please notice that the argument that the frequency of precipitation is more useful than intensity for understanding wet deposition lifetime changes in context of changes in aerosol lifetime for the LGM/current was used in the Mahowald et al., 2011 paper in Quaternary Science Reviews, which might help your arguments here.*

We have added the citation in MS. "*Mahowald et al.* [2011] also discussed the importance of precipitation frequency in wet deposition based on simulations showing large removal rate of dust in precipitation events."

Mahowald, N., et al.: Model insight into glacial–interglacial paleodust records. Quat. Sci. Rev. 30.7-8, 832-854, 2011.

**Response to Referee #3**

*Hou et al. systematically investigate the effect of precipitation frequency and intensity on aerosol scavenging using a coarse resolution global model with a rather simplistic description of aerosol scavenging. The topic is especially interesting since the changes in precipitation characteristics (e.g. more extreme precipitation and possibly less drizzle) constitute an important contributor to the climate change signal. While the finding that the change of the black carbon lifetime in a changing climate might be dominated by changes in precipitation frequency and not in precipitation amount does not seem overly surprising in the light of the cited literature, this study nevertheless seems very interesting and useful to me, especially since to my knowledge this study represents the first attempt to investigate the topic in such a focused and systematic fashion. The study nicely explains why an increase in total precipitation amount does not necessarily lead to a decrease in aerosol lifetime (independent of changes in spatial patterns that may for example impact some regions with high emissions more than others). In my opinion the manuscript serves to highlight a rather interesting and important topic and in spite of some limitations it can serve as a very good base for further studies. I recommend to publish the manuscript subject to minor revisions.*

*Specific comments/suggestions*

*1. l. 86ff: I think that (a) scaling the precipitation by a uniform factor for each grid box and (b) using a stochastic function where precipitation is turned off regardless of whether it is heavy or light precipitation may in principle lead to different outcomes compared to what might be expected from climate change (in which e.g. strong precipitation intensity may be enhanced while weak precipitation may decrease or remain unchanged and models also suggest very distinct spatial patterns) and I am not sure that the results from these very idealized sensitivity tests can be used to deduce a quantitatively correct answer for the climate change signal. I suggest to discuss this point. Also, as far as possible, I would appreciate if the authors could put their estimate of the change in aerosol lifetime into the context of other estimates from the literature, e.g. in the conclusion section in l. 241, although most of the existing literature estimates will not be directly comparable since they look at different regions and times. For example, Fang et al. (2011) estimate a change in lifetime for their SAt tracer. I also wonder if it would make sense to construct*

*Fig. 2 for each region separately?*

Very good points. We have added more discussion in the MS – "We should note that there are some caveats for our idealized sensitivity simulations. The way we reduce precipitation frequency in the model (based on a stochastic function as discussed in section 2) can be very different from climate-driven precipitation change in the real world. The globally uniform scaling factors applied to precipitation intensity do not account for the spatial variations. As a consequence, the sensitivities of BC lifetime to precipitation changes over a specific region may be different from those shown in Fig. 2."

In addition, we have constructed contour plots similar to Fig.2 but for various regions. It appears these regional plots are very close to the global one (Figure 6), so we decided to keep the global contour in Fig. 2-4 but add more analysis and discussion about the regional plots in the MS - "To partly address this issue, we have constructed some regional contour plots similar to that in Fig. 2 but based on sensitivities of BC lifetime for those specific regions (Fig. 6). Comparison of these regional contours with the global one indicate some differences in the sensitivity of BC to precipitation changes, but generally less than 3%."

We have added the comparison of our results and literature that "Our results are consistent with *Kloster et al.* [2010] and *Fang et al.* [2011] who reported increasing atmospheric aerosol burden due to climate change, although their results are based on future climate change."

[Figure]

**Figure 6: Compare the contours calculated on the global and regional scale: a). global; b). southeast North America (seNA); c). northeast Asia (neAS). The contours indicate the atmospheric lifetimes of the black carbon aerosols from the interpolation of 20 cases and show the potential changes of BC lifetimes from the base BC lifetime (in the control run) driven by the changes of precipitation intensity and frequency. The contour calculated on the global scale is the same with Fig. 2. seNA and neAS are two most extreme cases among all regions, with the smallest and largest sensitivities between BC lifetimes and precipitation changes.**

*2. Especially the time period covered by the TRMM dataset is rather short, so that influences of internal variability are likely to play some role at least on a regional bases.*

*On the other hand, the increase in precipitation intensity is consistent with expectations in a warming climate. It would also be interesting to see what part of the changes in precipitation frequency in Fig. 4 may be associated with internal variability, although I realize that this is outside the scope of this study. I think it would nevertheless be good to more explicitly mention that some of the regional trends may at least in part be due to internal variability e.g. in line 221. For example CMIP5 model simulations suggest that the effect of internal variability even on multi-decadal regional precipitation trends can be rather large, especially for small regions. The global average changes, on the other hand, are much more directly related to the forcing strength. The large spread in the values of precipitation frequency in Fig. 4 may also be an indication of internal variability, although I am not sure if one can obtain an estimate based on the existing literature. Further research which is outside the scope of this work may be required to quantify this. One way to "filter out" the effect of internal variability might be to compute the average change in the BC lifetime over all regions, although one could argue that this also means loosing other information that is contained in the regional averages (e.g. differences due to different characteristics of the regions) and that the regional lifetimes are generally of larger interest than the global average. My recommendation would nevertheless be to compute the 30-year changes of the global average BC lifetime for all the land areas (with the contributions from the individual regions weighted by the size of the individual regions) and also for the entire globe and to state the values in the conclusion section. This may then also facilitate a more meaningful discussion of the results from this study in relation to existing literature.*

We have shown the standard errors for the calculated changes in precipitation characteristics in Fig. 4 with the size of red block, which reflects the magnitude of the interannual variability in these precipitation fields. Covering a shorter time period, TRMM (Fig. 4) shows much larger standard errors than other datasets (Fig. 5). Based on the TRMM data, it appears the average changes in precipitation characteristics over this 14-year period are significantly larger than the interannual variabilities. To clarify this point, we have added more discussion in MS – "The changes in the average precipitation intensities and frequencies between the periods of 2008-2014 and 2001-2007 for each region are shown as ratios in Fig. 4, with the width and height of the blocks indicting the standard errors of the calculated percentage changes in precipitation frequency and intensity, respectively. Although these TRMM data only cover 14 years, the standard errors as shown in Figure 4 indicate that the changes in precipitation intensity and frequency over most regions are statistically significant."

*3. The parameter range that is explored in Fig. 2 seems very large in the context of global climate change and there seem to be relatively few sensitivity simulations that are in the range of expected climate change. On the other hand, any potential bias that results from this will most likely not be overly large in the light of other uncertainties that*

*stem from incomplete knowledge of actual and expected precipitation changes, uncertainties in the scavenging formulation, and possibly also uncertainties related to the design of the study (see my point one #1 above).*

Good point. We have added clarification in the MS – "In addition, to clearly demonstrate that the BC lifetime has different sensitivities to precipitation intensity and frequency, our sensitivity simulations cover a wide range of precipitation intensity and frequency. Some of these applied perturbations are significantly larger than those induced by climate change, especially at large (such as regional or global) scales. Therefore, simple interpolation of some results from this study in examining the effects from climate change may introduce some uncertainties."

*Other specific comments/suggestions/questions:*

*1. In the introduction, there are a few cases (Salzmann et al., line 28; Trenberth et al., 2007, line 29; Trenberth et al., 2011, line 31; Dawson et al., 2007, line 36; Fang et al., 2011, line 40) in which it might be nice to know what the cited findings are based on (e.g. observations, regional/global model, theoretical arguments, combination of modeling and observations, models constrained by observations, etc).*

We have added more information about the findings we cited. –

"*Salzmann* [2016] found that the global mean precipitation did not change significantly since 1850 with climate models, while *Trenberth et al.* [2007] reported that the total precipitation amount increased over land north of 30°N in the past century and decreased in the tropical region after the 1970s based on observational data. *Trenberth* [2011] also noted that theoretically a warmer climate could lead to less frequent but more intense precipitation. "

"For example, *Dawson et al.* [2007] found a strong sensitivity of the concentrations of the PM2.5 (particulate matter with diameter less than 2.5 μm) to precipitation intensity over a large domain of the eastern US with perturbation tests. Only a few studies focused on precipitation frequency."

"*Fang et al.* [2011] projected with the Geophysical Fluid Dynamics Laboratory chemistry-climate model (AM3) that wet deposition has a stronger spatial correlation with precipitation frequency than intensity over the US in January, although they concluded that frequency has a minor effect on wet deposition in the context of climate change."

*2. l. 59: in addition to the URL, please also cite at least one paper that describes GEOS-Chem, even if it not exactly the version that is used here.*

We have added a citation [*Bey et al.*, 2001] to describe GEOS-Chem.

Bey, I., et al.: Global modeling of tropospheric chemistry with assimilated meteorology: Model description and evaluation. J. of Geophys. Res. Atmos., 106.D19, 23073-23095, 2001.

*3. l. 72: unit of P?*

We have added the unit of P, which is mm/h.

*4. l. 78: did the authors check whether the results are sensitive to this definition?*

We picked this definition because it has been widely used in the literature and haven't explored other definitions. This is an interesting point and we may revisit this issue when we carry out relevant analyses in the future. Since we use data from 4 different datasets, re-processing and analyzing all the data would take a long time.

*5. l. 115f: did the authors check whether the result is sensitive to this?*

We did not check the sensitivity to grid resolution with TRMM data but based on the literature studies [e.g. *Huffman et al*., 2007; *Gehne et al*., 2016], the 2x2.5 resolution would help reduce the relative errors at small precipitation rates. Also the 2x2.5 resolution works well for the continental scale we are looking at in this study.

*6. l. 157 and lines 165ff: good points that are nicely explained.*

Thank you.

*7. l. 178: are those the standard deviations of the yearly means?*

No, we calculated the standard errors of the percentage changes using all the data points directly. The temporal resolution for the data varies across datasets; e.g. the TRMM data are 3-hr averages, while the NCEP data are 6-hr averages.

*8. l. 240 ff: "precipitation changes" is used here and also further below. It would be better to be more specific regarding whether this is mostly frequency or intensity.*

We have changed "precipitation change" to more specific "the changes of precipitation intensity and frequency".

*9. l. 251: "feedbacks" are usually understood to be mediated by sea surface temperature (SST) change. In a model run in which SSTs are prescribed based on observations, the effect of aerosol on SST during this period is actually taken into account. But the authors are right in the sense that assessing the magnitude of the feedbacks is not possible in such a setup.*

Good point. We have removed this sentence to avoid the possible confusion.

*Suggestions for technical corrections*

*l. 15: omit "simulation" l. 19: aerosols -> aerosol l. 26: other atmospheric elements -> soluble trace gases l. 67: details -> detail l. 86: control -> the control l. 93: simulation tests -> sensitivity tests l. 98: rate -> rates l. 104: precipitations -> precipitation l. 108: We -> . We l. 126 control -> the control l. 149: that -> that this l. 200: same -> the same l. 232: have -> has l. 315: year? l. 346: control -> the control*

Thank you very much for catching these. We have implemented all of these corrections in the MS.

*Fig 1: please increase the size of the labels (and/or magnify the figure) and increase the resolution so that the figure can be magnified on the screen. Please also consider increasing the resolution of Fig. 5.*

We have increased the size of the labels as well as the resolutions of figures and hope the high-resolution figures will carry over through the file uploading process.